# Simultaneous Impact of Rhizobacteria Inoculation and Leaf-Chewing Insect Herbivory on Essential Oil Production and VOC Emissions in *Ocimum basilicum*

**DOI:** 10.3390/plants13070932

**Published:** 2024-03-23

**Authors:** Tamara Belén Palermo, Lorena del Rosario Cappellari, Jimena Sofía Palermo, Walter Giordano, Erika Banchio

**Affiliations:** INBIAS Instituto de Biotecnología Ambiental y Salud (CONICET—Universidad Nacional de Río Cuarto), Campus Universitario, Río Cuarto 5800, Argentinalcappellari@exa.unrc.edu.ar (L.d.R.C.);

**Keywords:** essential oil, herbivory, plant growth-promoting bacteria, rhizobacteria, secondary metabolites, *Spodoptera frugiperda*, sweet basil, trichome density, volatile organic compounds

## Abstract

Inoculation with rhizobacteria and feeding by herbivores, two types of abiotic stress, have been shown to increase the production of secondary metabolites in plants as part of the defense response. This study explored the simultaneous effects of inoculation with *Bacillus amyloliquefaciens* GB03 (a PGPR species) and herbivory by third-instar *Spodoptera frugiperda* larvae on essential oil (EO) yield and volatile organic compound (VOC) emissions in *Ocimum basilicum* plants. The density of glandular trichomes was also examined, given that they are linked to EO production and VOC emission. Herbivory increased EO content, but inoculation on its own did not. When combined, however, the two treatments led to a 10-fold rise in EO content with respect to non-inoculated plants. VOC emissions did not significantly differ between inoculated and non-inoculated plants, but they doubled in plants chewed by the larvae with respect to their undamaged counterparts. Interestingly, no changes were observed in VOC emissions when the treatments were tested together. In short, the two biotic stressors elicited differing plant defense responses, mainly when EO was concerned. PGPR did not stimulate EO production, while herbivory significantly enhanced it and increased VOC emissions. The combined treatment acted synergistically, and in this case, PGPR inoculation may have had a priming effect that amplified plant response to herbivory. Peltate trichome density was higher in inoculated plants, those damaged by larvae, and those subjected to the combination of both treatments. The findings highlight the intricate nature of plant defense mechanisms against various stressors and hint at a potential strategy to produce essential oil through the combined application of the two stressors tested here.

## 1. Introduction

Medicinal and aromatic plants include a wide variety of species that produce essential oils (EOs). Sweet basil (*Ocimum basilicum* L.), a prominent aromatic plant from the Lamiaceae family, is used both fresh and dry as a condiment, and its extracts are essential for the production of pharmaceuticals, cosmetics, and perfumes, among other industrial goods [1]. The main compounds in sweet basil EO are terpenes and phenylpropanoids, followed by alcohols and aldehydes [2]. This composition may vary depending on factors such as the plant’s geographical origin, its variety, its developmental stage, and the agricultural conditions of its cultivation [3]. Components commonly cited in the literature include 1,8-cineole, linalool, camphor, eugenol, methyl eugenol, methyl cinnamate, methyl chavicol, and germacrene [4,5]. According to several studies [6], the phenolic compounds in sweet basil EO are responsible for many of the antimicrobial, antifungal, insect-repelling, antioxidant, anticancer, and anti-inflammatory properties ascribed to it [1,7,8,9,10].

Plants must discriminate between different environmental conditions and respond to each appropriately to distribute resources effectively for growth, reproduction, and defense [11]. Attack by pathogens or insects, for instance, is an adverse condition that triggers a series of defense mechanisms whose purpose is to stop, lessen, or counteract the damage [12,13,14]. However, plants also benefit from positive interactions with beneficial microorganisms. Plant growth-promoting rhizobacteria (PGPR) are capable of colonizing the plant root system and enhancing plant growth and performance through direct and indirect mechanisms [15]. The former involves the stimulation of plant development through the production of growth regulators (auxins, cytokinins, gibberellins, abscisic acid), biological nitrogen fixation, and the solubilization and mineralization of phosphates [16,17]. The latter includes the synthesis of antibiotics, antifungals, or siderophores and the induction of the plant’s defense responses (induced systemic resistance or ISR) to inhibit or fight off phytopathogenic microorganisms [18,19].

Plant defense responses are regulated by activating phytohormone-controlled signaling pathways [20,21]. Attacks by insects also induce ISR, which implicates signal transduction via phytohormonal pathways, changes in gene expression, and eventually, responses such as the biosynthesis of secondary metabolites [22,23,24]. Herbivore feeding induces the regulation of terpenoid and phenylpropanoid biosynthesis [25,26]. The rise in secondary metabolite concentrations, which occurs in response to stress, is often mediated by an increase in the transcriptional activity of specific biosynthetic genes [27]. This increase is controlled by a complex signaling cascade in which the hormone jasmonate plays an important role [28]. Exogenous treatment of *O. basilicum* plants with methyl jasmonate significantly increased their EO yield by modifying the regulation of terpene synthase genes [29,30]. Similarly, sweet basil treated with different concentrations of jasmonic acid showed increased EO yield, particularly in linalool and eugenol levels [31]. This agrees with previous findings obtained with methyl jasmonate treatment [32,33].

Mechanisms involving jasmonic acid and ethylene are likewise deployed in the induction in ISR by root cells after they perceive rhizobacteria [34,35]. PGPR are able to stimulate the biosynthesis of secondary metabolites [36,37,38,39,40]. Direct inoculation of medicinal and aromatic plants (*Origanum majorana*, *Origanum x majoricum*, *Tagetes minuta*, *Mentha piperita,* and *O. basilicum)* with different PGPR species led to a significant increase in plant development and the production of secondary metabolites such as EO and phenolic compounds [38,41]. Importantly, plant responses to PGPR have been observed to vary in degree between one plant species and the next, which demonstrates the specificity of the microorganism–plant relationship [42]. Studies in which sweet basil was inoculated with PGPR found an increase in most plant growth parameters, such as fresh/dry shoot weight and leaf area index, as well as improved EO yield [43,44,45]. When inoculated with *B. amiloliquefaciens* GB03, *O. basilicum* exhibited enhanced growth and EO yield [46].

A critical dual role in plant defense is played by trichomes, which provide both structural and chemical protection [47]. *O. basilicum* features capitate and peltate glandular trichomes on both leaf surfaces. Peltate trichomes are involved in the synthesis and accumulation of EO; capitate trichomes are associated with the presence of polysaccharides. Due to their low molecular weight, low water solubility, and high vapor pressure, EOs can volatilize into the atmosphere. This release of volatile organic compounds (VOCs) is pivotal in plant interactions with pollinators, herbivores, and other plants. It significantly improves the defense against pests and pathogens by attracting natural pest enemies after herbivore damage [48].

As mentioned above, the positive effects of PGPR inoculation and herbivory on the production of secondary metabolites have been well documented by studies that looked into these treatments separately. However, their combined impact remains relatively unexplored. The present study aimed to determine how this combination affects the biosynthesis, accumulation, and emission of EOs in *O. basilicum*. By simultaneously examining the impact of rhizobacteria inoculation and leaf-chewing insect herbivory on secondary metabolites, we gained insight into the complex interactions between plants, rhizobacteria, and herbivores, which shape plant defense mechanisms.

## 2. Results

### 2.1. EO Content and Main Constituents

The chemical composition of the essential oil of *O. basilicum* L. is shown in Table 1. The constituents have been listed in ascending order according to their retention times.

EO content was not altered (*p* > 0.05) by inoculation with GB03 (Figure 1). However, it increased three-fold with respect to the control (*p* < 0.05) in plants that suffered damage by *Spodoptera frugiperda* larvae (SF). On the other hand, it increased tenfold when the plants were exposed to both treatments in combination, which demonstrates synergy between the treatments (Figure 1). This was further corroborated by the two-way analysis of variance, which showed a highly significant interaction between the effects of the two treatments on EO (*p* < 0.05). In other words, herbivory and inoculation have influenced each other, leading to a significant rise in EO content compared to each treatment.

The treatments were also associated with differences in the concentration of major components in the EO (Figure 2). The concentration of cineole and terpineol did change significantly after treatment in relation to the control (*p* > 0.05). However, the combination of both treatments resulted in a 6- and 17-fold increase in these components, respectively, compared with control plants (*p* < 0.05). Linalool and eugenol were approximately 2–4-times higher in plants that received the individual treatments than in the control group (*p* > 0.05). When plants were inoculated and subsequently damaged by SF, the increase in these components was much more remarkable, with levels being 9–13-times higher than in the control (*p* < 0.05).

### 2.2. Emission of Plant VOCs

The main compounds released by sweet basil plants were linalool, cineole, eugenol, and terpineol. The statistical analysis revealed that the emission of VOCs by inoculated plants was not significantly different from the non-inoculated control group (*p* > 0.05) (Figure 3). In contrast, VOC emissions increased three-fold with respect to the control in plants that were damaged by SF (*p* < 0.05).

The emission of volatiles by plants that received the combination of treatments was lower than by those plants that were solely damaged by the herbivore (*p* < 0.05) (Figure 3). Variations were recorded in the emission of each compound depending on the treatment, except in the case of cineole, which remained unaltered (Figure 4). The emission of linalool and eugenol was approximately two-times higher in plants exposed to SF than in the control (*p* < 0.05). Inoculation applied on its own resulted in an increase in eugenol emissions and a decrease in linalool emissions (*p* < 0.05) with respect to control plants. The combined treatment only caused a significant variation in linalool emissions, which were 50% higher than in the control.

After conducting a multivariate analysis, the data was used to create a clustered heat map on ClustVis, a web-based tool. This map (Figure 5) made it possible to contrast the effects of the two treatments and their combination on the main compounds in the EO and the emitted VOCs. As seen in Figure 5A, which shows the EO composition, the most intense area (red) corresponds to eugenol, i.e., the concentration of this compound in the EO was more significant than that of other compounds. Nevertheless, in Figure 5B, which represents the emitted VOCs, linalool appears red, and eugenol is blue. This indicates that the concentration of key compounds in the EO was not always directly proportional to their emitted levels, regardless of whether the plants received the treatments separately or in combination.

### 2.3. Trichome Density

Histological units called glandular trichomes are responsible for producing and storing EO. Although these trichomes were found on both sides of the leaf in *O. basilicum*, they were more abundant on the abaxial side (Figure 6). The density of capitate trichomes (CT) increased on both sides only in the leaves of plants that were inoculated and damaged by herbivory (*p* < 0.05). Peltate trichomes (PT), the primary site for EO synthesis [49], were denser in inoculated than in non-inoculated plants. More precisely, their density was around 50% higher on the abaxial side and 20% higher on the adaxial side than in the control. When comparing plants exposed to herbivory to their control, the increase in PT density was more noticeable: the trichomes were 70% and 40% denser on the abaxial and adaxial side, respectively (*p* < 0.05). The combination of treatments, however, did not lead to significant differences in PT density with respect to the corresponding controls (exposure to SF larvae or GB03 inoculation), on either side of the leaf (*p* > 0.05).

### 2.4. Principal Component Analysis

A multivariate principal component analysis (PCA) was performed to establish relationships between the treatments tested (inoculation, herbivory, and their combination) and the factors evaluated (peltate trichome density, EO yield, and VOC emissions). This type of analysis renders a graph (Figure 7) that facilitates the visualization and interpretation of the data set and the variables. As regards the treatments, the analysis revealed that CP1 (herbivory) explains 70.4% of the variability in the data, while CP2 (inoculation) is responsible for 25.3%. Together, both axes explain 95.7% of the variations in the data and have a cophenetic correlation coefficient of 0.997. There is a strong positive correlation (acute angle) between peltate trichome density, on the one hand, and EO content and VOC emissions, on the other. The angle between EO yield and VOC emissions is less pronounced, which means their correlation is weaker. The plot shows that plants damaged by SF are near almost all the variables evaluated (blue circle). More specifically, they are closer to EO yield after having received combined treatment and closer to VOC emissions when exposed to the larvae without previous inoculation. On the other hand, plants not damaged by SF were far from the variables evaluated, regardless of whether they were inoculated (red circle), which indicates a low impact.

## 3. Discussion

The rhizosphere and its associated microbiome are key drivers of crop health and productivity [50]. According to earlier studies carried out by our research group and other authors on species other than sweet basil, colonization by PGPR creates optimal growth conditions for medicinal and aromatic plants and is associated with increased total fresh weight, number of leaves, stem length, and root dry weight [36,38,46,51]. Additionally, PGPR can trigger ISR, a critical response when mediated by the bacteria in conjunction with herbivory. PGPR activates ISR by stimulating the jasmonic acid and ethylene signaling pathways and, in so doing, enhances the plant’s capability to ward off pathogens and insect pests [17].

When a plant undergoes biotic stress, ISR can mediate morphological, physiological, and molecular alterations in plant tissues. These changes can lead to modifications in the type, composition, and concentration of phytochemicals, which are also responsible for plant defense. By increasing its production of phytochemicals, the plant can become more resistant to future attacks [52,53].

In the present study, no differences were found in EO content between sweet basil plants inoculated with *B. amyloliquefaciens* GB03 and non-inoculated plants. This contrasts with previous studies, where an increase in EO content was recorded in different plant species after PGPR inoculation [39,40], particularly in aromatic plants such as *Mentha piperita* [37], *O. majoricum*, *O. majorana* [36,51], and *T. minuta* [38].

When the basil plants were exposed to herbivory by *S. frugiperda*, an increase in the content of total EO and its major compounds was observed. This is likely part of the plant’s defensive response to herbivore attacks, considering that specific compounds in sweet basil EO are known to have antifeedant and inhibitory activity [54]. Eugenol, for instance, has demonstrated a remarkable inhibitory effect on α-amylase and total proteases in *S. littoralis* larvae, so it has the potential to disrupt digestive processes in herbivores [55]. Furthermore, eugenol has been found to inhibit the acetylcholinesterase (AChE) enzyme in *S. frugiperda*, which is further evidence of its larvicidal and antifeedant properties [56]. Linalool, another compound in *O. basilicum* EO, interacts with the cholinergic system of insects [57] and might also modulate AChE [58]. It is a valuable component in the plant’s defense against *S. frugiperda* due to its broad spectrum of toxicity, which includes acute toxicity, repellent action, and a knockdown effect [59]. In agreement with our results, Agliasa and Maffei [60] recorded an increase in the total content of terpenes and sesquiterpenes in *O. vulgaris* plants damaged by *S. littoralis*. The terpineol, limonene, and linalool content were higher than in control plants. Similarly, Cappellari et al. [61] reported an increase in EO yield in *M. piperita* damaged by *Rachiplusia nu* larvae, both in inoculated and non-inoculated plants. In addition, mechanical damage and leaf punctures produced by *Liriomyza huidobrensis* induced changes in the EO composition of *Minthostachys mollis* [62,63].

On the other hand, VOCs have a significant impact on ecological functions and can alter behavior and physiology in a wide range of organisms [64]. Plants damaged by herbivores emit volatiles, which are necessary signals for parasitoids and predators to locate their hosts [48]. In the present study, VOC emissions from inoculated sweet basil plants did not vary with respect to those by control plants. This differs from a previous study on *M. piperita*, which registered an increase in VOCs after inoculation with GB03 or co-inoculation with GB03 and *Pseudomonas putida* SJ04 [65]. An increase in total VOCs was also observed in other non-aromatic crops, such as maize inoculated with *B. thuringiensis* RZ2MS9 or co-inoculated with RZ2MS9 and *Azospirillum brasilense* Ab-v5 [66].

Nevertheless, *S. frugiperada* larvae damaged our plants, there was a 2.2- and 2.75-fold increase in the emissions of eugenol and linalool, respectively. In contrast, dos Santos Tozin et al. [67] reported that after exposure to leaf-cutting ants, eugenol in O. gratissimum plants decreased by 16 to 5% and terpineol by 0.36% to undetectable levels, although cineole levels remained unchanged. The disparity with our findings might be attributed to the differences in feeding behavior between the insects used (chewing and cutting), and the fact that different herbivore species could induce different plant responses. Along with tissue damage, herbivores trigger defense responses via effectors. These effectors can have diverse structures, e.g., enzymes, modified forms of lipids, sulfur-containing amino acids, or peptides released from digested plant protein [68]. This highlights the intricate nature of VOC emissions and shows that specific types of inflicted damage play a crucial role in shaping specific plant responses.

When *O. basilicum* plants were inoculated with GB03 and damaged by *S. frugiperda* larvae, EO levels increased 6- and 3-fold with respect to the controls, which had been either solely inoculated or exposed to the larvae. The combination of treatments, therefore, appears to have acted synergistically to elicit a more significant defensive response. These results diverge from a study in which the rise in EO content did not differ significantly between *M. piperita* plants that were inoculated with PGPR, damaged by *R. nu,* or subjected to the two treatments [61]. The variability in the results highlights the specificity of the effects of PGPR–plant interactions on defense responses from one species to another [34]. Our observations regarding the synergy of the combined treatment in *O. basilicum* could be attributed to a priming effect by PGPR. In other words, initial exposure to the bacteria may have improved the plants’ response to subsequent exposure to the larvae. During priming, plants become more sensitive to signaling hormones such as jasmonic acid or ethylene, enabling them to reprogram their metabolome better to defend themselves against future stressful events [53,69]. When exposed to stress, primed plants produce defense metabolites earlier and in larger quantities than unprimed plants. Such metabolites include phenylpropanoids, terpenoids, polyketides, and alkaloids [53,69,70], as observed in the present study. As mentioned earlier, an increase in the production of secondary metabolites has been suggested to be part of the defensive response in plants since the active compounds in EOs can negatively affect several pests [71]. In previous studies, the inoculation of sweet basil with GB03 caused changes in plant phytochemistry, which affected the development of *S. frugiperda*. The secondary metabolites synthesized by the inoculated plants effectively delayed *S. frugiperda* development, reduced the size of the pupa, and decreased the number of insects transitioning from the pupal stages to their adult form. All of this can reduce the insect population, making them more susceptible to diseases and natural enemies [72]. The negative effects of *O. basilicum* EO have also been reported on other insects, with specific compounds playing a crucial role [73].

The increase in EO content in plants that received the combination of treatments in our study was not associated with denser glandular trichomes. This is particularly significant, considering the well-established correlation between the content of secondary metabolites and trichome density [74]. However, results analogous to ours (i.e., no correlation between the variables) were obtained in *Stevia rebaudiana* plants inoculated with endophytic bacteria [75].

Given that the combination of treatments acted synergistically on EO content, we expected a similar response regarding VOCs. However, VOC emissions in plants inoculated and subsequently exposed to herbivory were not higher than in the control. Moreover, while the chemical composition of the VOCs shares similarities with that of the EO, there are notable differences in the major components, as illustrated by the heatmap (Figure 4). Eugenol is predominant in the EO, but linalool primarily characterizes the emissions.

Other studies have observed similar discrepancies between VOC emissions and EOs in different plant species, in terms of overall content and major compound composition [76,77]. This phenomenon raises interesting questions about the regulation of the production and release of volatile compounds in plants. Much is known about the chemical structures of plant volatiles, the pathways, enzymes, and genes underlying their biosynthesis, and the factors regulating their formation. However, more information is needed about the last step that occurs in the plant, i.e., how volatiles are released into the atmosphere. In aromatic plants, volatiles are emitted from glandular trichomes, where they are stored, and healthy plants typically maintain a basal emission of these volatiles [78]. Recent research challenges the assumption that volatile compounds move solely through passive diffusion. High barrier resistance across cellular components, including the cytosol, the plasma membrane, the aqueous cell wall, and the cuticles for lipophilic VOCs, contradicts the idea that diffusion alone explains the high emission rates registered during stress responses. Fick’s first law of diffusion suggests VOCs accumulate internally until reaching toxic levels before they are emitted, which means more active trafficking mechanisms might be involved in their release. Some possibilities include vesicular trafficking, soluble carrier proteins, ABC transporters, and small carrier proteins like lipid transfer proteins (LTPs) [79,80,81]. Notably, the ABC transporter PhABCG1 has been identified as a transporter of benzenoid and phenylpropanoid compounds in petunia [82]. This hints at the sophisticated system that plants may use for more precise communication through volatile-encoded messages.

## 4. Materials and Methods

### 4.1. Bacterial Strains, Culture Conditions and Media

The PGPR strain used in this study was *Bacillus amyloliquefaciens* GB03. It was cultured on LB medium and preserved in nutrient broth with 15% glycerol at −80 °C for long-term storage. For experimental purposes, the bacteria were cultured on nutrient agar. Single colonies were then transferred to 100 mL flasks containing the appropriate culture medium and grown aerobically on a rotating shaker (150 rpm) for 48 h at 28 °C. The resulting bacterial suspension was diluted in a sterile saline solution (0.9% sodium chloride, NaCl) to achieve a final concentration of 10^8^ colony-forming units (CFU) per milliliter. Subsequently, 1000 μL of this suspension was applied as an inoculum around the base of the plant stems.

### 4.2. Insects

Spodoptera frugiperda (SF) larvae were provided by AgIdea (Agricultural Innovation Applied Research, Pergamino, Argentina). They were obtained from a colony without pesticide exposure and kept on a semi-synthetic diet [83] at 23–25 °C in a 70% humidified chamber, with a 16:8 h light/dark photoperiod.

### 4.3. Seed Sterilization and Plant Cultivation

Seeds of *Ocimum basilicum* L. var. genovesa (Florensa Argentina S.A) were surface-sterilized by soaking for 2 min in 70% (*v*/*v*) ethanol and for 20 min in 1% (*v*/*v*) sodium hypochlorite. After this, they were thoroughly rinsed four times with sterile distilled water and placed in plastic pots filled with sterilized vermiculite. Following a 15-day period, the plantlets were transplanted into larger plastic pots (12 cm × 22 cm) filled with sterilized vermiculite. They were grown in a growth chamber under controlled conditions of light (16/8 h light/dark cycle), temperature (22 ± 2 °C), and relative humidity (~70%) and watered every week with 20 mL of Hoagland solution per pot. After seven days, the plants were inoculated with 1000 μL of bacterial suspension or with sterile water in the case of control plants. The experiments were conducted three times (10 pots per treatment, 1 plant per pot), and arranged randomly in the growth chamber.

### 4.4. Bioassays and Treatments

Forty-five days post-inoculation, each plant was exposed for 4 h to three previously starved *S. frugiperda* larvae. This test was conducted within entomological cages. Forty-eight hours post-damage, VOCs were assessed, and the material was harvested, weighed, and transferred to liquid nitrogen for subsequent analysis. Several studies have revealed changes in secondary metabolites 48 h after herbivory [84,85].

The treatments were: (a) control; (b) plants inoculated with *B. amyloliquefaciens* GB03 (GB03); (c) plants infested with *S. frugiperda* (larvae); and (d) plants inoculated with *B. amyloliquefaciens* GB03 and infested with *S. frugiperda* (GB03+ larvae).

### 4.5. EO Extraction

Shoot samples were individually weighed and subjected to hydrodistillation in a Clevenger-like apparatus for 20 min. The volatile fraction was collected in dichloromethane, and p-cymene (2 μL in 400 μL of dichloromethane) was added as an internal standard since we had ascertained in earlier studies that it is not present in our basil plants [51]. *O. basilicum* plants yield ~3% EO, consisting of >20 different compounds. Its major components, which comprise ~70% of the total oil volume, are 1,8-cineole, eugenol, terpineol, and linalool. These compounds were quantified in relation to the standard added during the distillation procedure, as described above. Flame ionization detector (FID) response factors for each compound generated essentially equivalent areas (differences < 5%).

Chemical analyses were performed in a Perkin-Elmer Q-700 gas chromatograph (GC) equipped with a CBP-1 capillary column (30 m × 0.25 mm, film thickness 0.25 μm), and a mass selective detector. The analytical conditions were as follows: injector temperature 250 °C, detector temperature 270 °C; oven temperature programmed from 60 °C (3 min) to 240 °C at 4°/min; carrier gas: helium at a constant flow rate of 0.9 mL/min; source 70 eV. The oil components were identified based on mass spectral and retention time data compared to standard compounds [86]. The GC analysis was performed using a Trace 1300 GC Thermo Fisher Scientific gas chromatograph fitted with a TG-capillary column 5MS (30 m × 0.25 mm, 0.25 µm). The GC operating conditions were as follows: injector and detector temperature 250 °C; oven temperature programmed from 60 °C (3 min) to 240 °C at 4°/min; detector: FID; carrier gas: nitrogen at a constant flow rate of 0.9 mL/min.

### 4.6. Collection of Plant VOCs

The collection system consisted of a vacuum pump that created a constant airflow (300 mL/min) through a polyethylene terephthalate (PET) chamber (volume 1000 mL) containing a plant. This chamber was closed at one end with a cap that had been pre-drilled to fit the collection trap. At the other end, there was a cap with a hole through which the plant stem passed, and this separated the bottom of the chamber from the base of the pot. Air exited the chamber through a usable glass collection trap packed with 30 mg Super-Q adsorbent (80–100 mesh; Alltech, Nicholasville, KY, USA), which was rinsed 5 times with 10 mL dichloromethane before each collection to remove impurities. Headspace VOCs were collected for 2 h and eluted immediately from the absorbent traps with 200 mL dichloromethane, after which the internal standard was added (1 μL p-cymene in 50 μL dichloromethane) [63]. Collected VOCs were analyzed by gas chromatography as described above. Following VOC collection, each plant was cut and weighed, with VOCs collected from control (uninoculated) plants. Collections from an empty chamber showed that the background level of monoterpenes was negligible.

### 4.7. Trichome Density

To study glandular trichomes, the plants were kept under controlled conditions in a growth chamber following insect exposure until new fully expanded leaves appeared 30 days after the damage [67].

A layer of acrylic was coated onto both sides of the leaves, then meticulously removed and mounted for microscopy on a solution of glycerol/distilled water (1:10) [87]. Six leaf blades were processed for each treatment. Trichome density (number/mm^2^) was determined from three microscope fields chosen at random from each leaf epidermis. Histological preparations were evaluated with a standard Zeiss model microscope. Photographs were taken with a Zeiss Axiophot microscope equipped with image capture and digitization (Ds-Qi1Mc, Nikon Eclipse 50i). Peltate and capitate trichomes were counted on both sides of the leaves using records from 5 microscopic fields chosen randomly and observed at a magnification of 10×. Image analysis was performed on Microsoft Paint for Windows.

### 4.8. Statistical Analyses

The data’s normality and homogeneity of variance were initially assessed using Shapiro–Wilk and Levene tests, respectively. Due to the observed lack of homogeneity, a linear mixed-effects model (MLM) was used. Comparisons between variables measured under the different treatments (EO content, VOCs, and trichome density) were made with MLM using analysis of variance (ANOVA). Models with homogeneous and heterogeneous residual variances were compared through the likelihood ratio test. The best-fitting statistical model included the fixed effects of inoculation, insects, and inoculation-insects, with heterogeneous variances in inoculation-insects. After MLM, means were compared with Fisher’s LSD test (α = 0.05). The cluster heatmap was generated using the web-based tool ClustVis [88], using Euclidean distance as the similarity measure and hierarchical clustering. Principal component analysis (PCA) was conducted to extract and display relationships between factors in the multivariate data set (herbivory and inoculation, trichome density, EO yield, and total VOC emission). Differences between means were considered significant for *p* values < 0.05. All statistical analyses were performed on Infostat v. 2020 (Infostat, Universidad Nacional de Córdoba, Córdoba, Argentina).

## 5. Conclusions

Two biotic stress types, PGPR inoculation and herbivory, elicited differing defensive responses in O. basilicum plants, particularly concerning EO. PGPR did not increase EO content, but leaf-chewing herbivory enhanced it and raised VOC emissions. The combined application of the two treatments had a synergistic effect on EO production, which suggests that PGPR inoculation may have primed the plants for a better response against the insects. The results are evidence of the intricate nature of plant defense mechanisms against various stressors and uncover significant possibilities for enhancing EO productivity, especially in sweet basil. High EO yields might be obtained through strategic inoculation with GB03 and controlled exposure to herbivory, preferably without the application of insecticides unless the damage is substantial. However, rigorously designed trials must be conducted in the field before considering the practical implementation of such a strategy. Future studies with inoculated plants should accurately measure the extent of damage caused by herbivory to determine the threshold at which it can be favorable for EO production.

## Figures and Tables

**Figure 1 plants-13-00932-f001:**
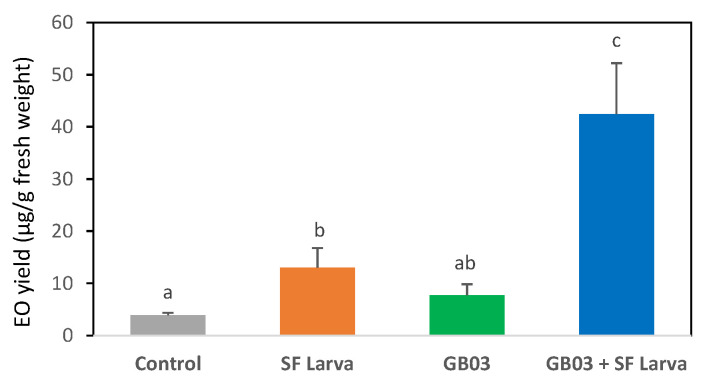
Essential oil yield in *Ocimum basilicum* plants exposed to herbivory by *Spodoptera frugiperda* larvae herbivory and/or inoculated with *Bacillus amyloliquefaciens* GB03 (mean ± SE). Letters above bars indicate significant differences according to Fisher’s LSD test (*p* < 0.05).

**Figure 2 plants-13-00932-f002:**
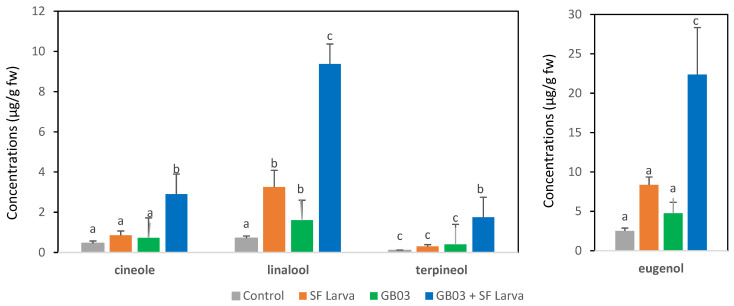
Concentrations of major EO components (μg/g fw) in *O. basilicum* inoculated with PGPR and/or infested by *S. frugiperda.* Letters above bars indicate significant differences according to Fisher’s LSD test (*p* < 0.05).

**Figure 3 plants-13-00932-f003:**
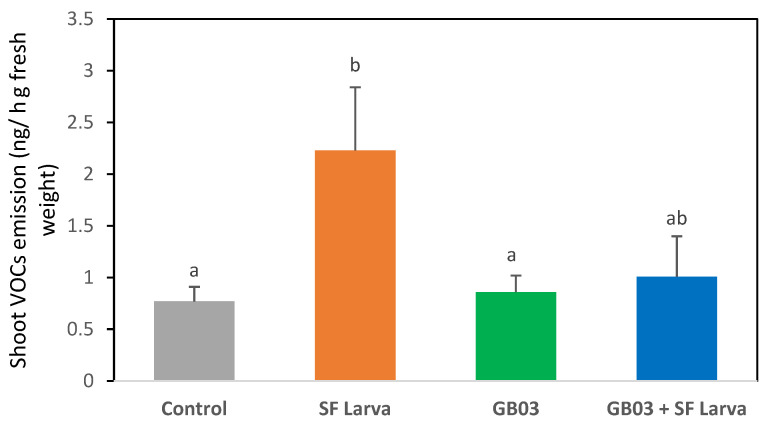
Emission of volatile organic compounds (VOCs) by *O. basilicum* plants inoculated with *B. amyloliquefaciens* (GB03) and/or exposed to *S. frugiperda*. Different letters indicate statistically significant differences according to Fisher’s LSD test (*p* < 0.05).

**Figure 4 plants-13-00932-f004:**
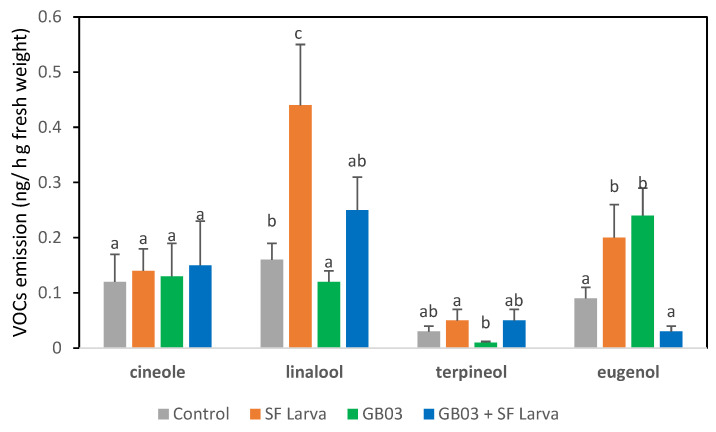
Concentrations of major VOC compounds emitted by *O. basilicum* plants inoculated with PGPR and/or infested by *S. frugiperda*. Different letters indicate statistically significant differences according to Fisher’s LSD test (*p* < 0.05).

**Figure 5 plants-13-00932-f005:**
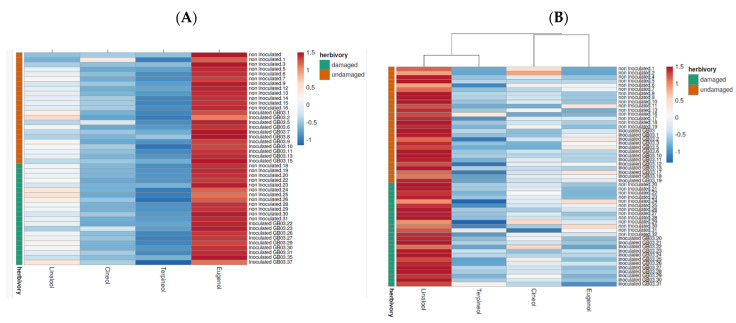
Clustered heat map of the effects of *Spodoptera frugiperda* herbivory, inoculation with *B. amiloliquefaciens* GB03, and their combination on: (**A**) EO composition, and (**B**) emitted VOCs.

**Figure 6 plants-13-00932-f006:**
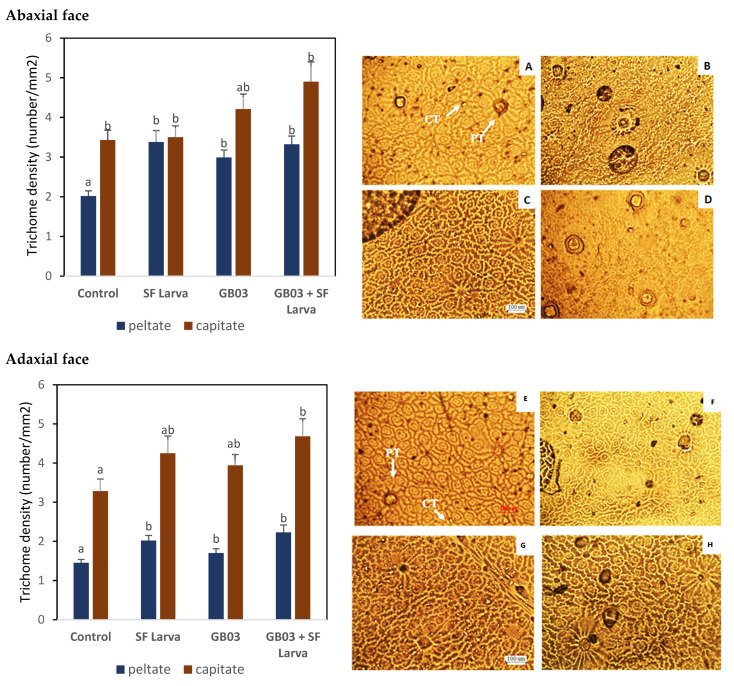
Effects of inoculation with *B. amiloliquefaciens* GB03 and/or exposure to SF on the density of peltate and capitate glandular trichomes in *O. basilicum* plants. (**A**,**E**) Control; (**B**,**F**) plants inoculated with GB03; (**C**,**G**) plants exposed to SF; and (**D**,**H**) plants inoculated with GB03 and exposed to SF. Values are mean ± standard error (SE). Means followed by the same letter in a given column are not significantly different according to Fisher’s LSD test (*p* < 0.05).

**Figure 7 plants-13-00932-f007:**
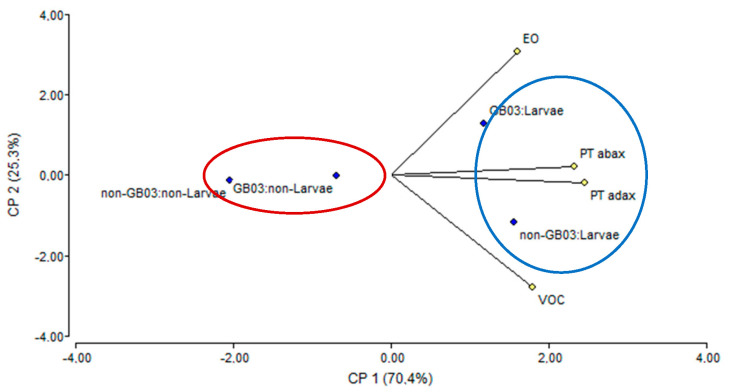
Principal component analysis for the physiological response of *O. basilicum* to insect chewing herbivory and PGPR inoculation. EO: essential oil yield, VOC: volatile organic compound emissions, PTadax: density of peltate trichomes on the adaxial side, and PTabax: density of peltate trichomes on the abaxial side.

**Table 1 plants-13-00932-t001:** Chemical composition of essential oil (%) from leaves of *Ocimum basilicum*.

Retention Time (min)	Components	Relative Percentage (%)
10.19	1,8-cineole	8.12
12.48	linalool	21.43
13.79	camphor	0.24
14.51	boreneol	0.57
15.27	α-Terpineol	1.26
18.08	bornyl acetate	0.70
20.034	eugenol	65.42
21.26	methyl eugenol	0.70
27.08	methyl cinnamate	2.25

## Data Availability

The data presented in this study are available on request from the corresponding author.

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
