# Peer review of "Simultaneous Impact of Rhizobacteria Inoculation and Leaf-Chewing Insect Herbivory on Essential Oil Production and VOC Emissions in *Ocimum basilicum"

_plants, 2024, doi:10.3390/plants13070932_

Round 1
Reviewer 1 Report
Comments and Suggestions for Authors
This study describes the increase essential oil and volatile organic compounds in Basil plants after inoculation with Rhizobium-type bacteria in combination with insect feeding by Spodoptera caterpillars. The authors found that Spodoptera feeding increased EO concentration but inoculation and feeding significantly increased EO production. However, VOC production increased with Spodoptera feeding, but seemed to be suppressed when combined with inoculation. The authors used appropriate statistical methods, including MLM with ANOVA and PCA, to analyze their data. Overall, the paper is scientifically sound but requires some edits before I feel it is fit for publication. My comments are below.
Lines 24-25 are confusing. Please reword the sentence (it is currently a sentence fragment).
Line 28 decisive is the incorrect word to use here. Maybe "Critical" or "important" are better choices
LIne 42 Can you think of a better way to say that plans "being sessile organisms"? It is trite and used in almost every paper ever written about plant defense responses.
Introduction- You need at least a paragraph in your introduction describing the differences between EOs and VOCs. You briefly talk about VOC emission in the discussion (lines 330-333), but this should be moved to the introduction. This will give the reader context about why each group is important and what each does.
Lines 96, 117, and 251. You mention multiple times that EOs after inoculation with GB03 "Increased 2 fold". You really can't say this. This could be due to your lack of a larger sample size, too much variation in your data, or may be just an artifact. There was no difference in EO levels between inoculated and controlled. To add that there was a 2 fold increase is misleading. Please remove references to this.
Tables 1, 2, and 3 are busy and would be more visually understandable and appealing if represented by a graph (Clustered column graph??)
Missing: Where is your list of the different EOs and VOCs that were found in your samples? I am seeing only 4 in Figure 3. Where these the only 4?
Line 279 This sentence is a fragment and should be reworded.
Line 282. You are dismissing the importance of specific insect salivary or feeding-method elicitors that may cause differences in EOs and VOCs in plants. 2-3 sentences with references should address this potential reason for the contrast between your results and studies done with ants.
LIne 303: What do you mean by decreased adult emergence? Please clarify what you mean.
313 Is "remarkable" the correct word choice?
316 "That not correlated" is unclear
Missing Please somewhere in the manuscript (discussion or intro) define the difference types of trichomes your were observing and the function of each.
Line 351. Where did you acquire the Gb03 culture?
Line 365 How did you separate instars?
LIne 383. When you did your insect bioassays, how did you ensure that insect feeding over 48 hours was consistent across your plants? Some insects may not have fed at all while others may have consumed the whole plant. Did you have a threshold for damage? Did you correlate damage on the plants with EO or VOC production?
Comments on the Quality of English Language
English language is mostly fine with minor grammatical errors.
Author Response
Reviewer #1:
Comments and Suggestions for Authors
Lines 24-25 are confusing. Please reword the sentence (it is currently a sentence fragment).
We apologize for the mistakes. The sentence has been reworded.
Line 28 decisive is the incorrect word to use here. Maybe "Critical" or "important" are better choices
The entire sentence has been rephrased.
LIne 42 Can you think of a better way to say that plans "being sessile organisms"? It is trite and used in almost every paper ever written about plant defense responses.
The phrase "being sessile organisms” has been removed
Introduction- You need at least a paragraph in your introduction describing the differences between EOs and VOCs. You briefly talk about VOC emission in the discussion (lines 330-333), but this should be moved to the introduction. This will give the reader context about why each group is important and what each does.
A paragraph detailing the differences between essential oils (EOs) and volatile organic compounds (VOCs) has been added to the introduction, providing necessary context on their importance and functions.
Lines 96, 117, and 251. You mention multiple times that EOs after inoculation with GB03 "Increased 2 fold". You really can't say this. This could be due to your lack of a larger sample size, too much variation in your data, or may be just an artifact. There was no difference in EO levels between inoculated and controlled. To add that there was a 2 fold increase is misleading. Please remove references to this.
We apologize for the errors and have removed the reference due to its lack of statistical significance.
Tables 1, 2, and 3 are busy and would be more visually understandable and appealing if represented by a graph (Clustered column graph??)
Tables 1, 2, and 3 have been replaced with clustered column graphs
Missing: Where is your list of the different EOs and VOCs that were found in your samples? I am seeing only 4 in Figure 3. Where these the only 4?
A table listing all EOs compounds identified in our samples has been added to the manuscript
Line 279 This sentence is a fragment and should be reworded.
The sentence in question has been reworded
Line 282. You are dismissing the importance of specific insect salivary or feeding-method elicitors that may cause differences in EOs and VOCs in plants. 2-3 sentences with references should address this potential reason for the contrast between your results and studies done with ants.
A sentence has been incorporated into the discussion to recognize the potential impact of distinct insect feeding behaviors on the variations in plant responses observed in our study compared to research involving ants,
LIne 303: What do you mean by decreased adult emergence? Please clarify what you mean.
The clarification regarding "decreased adult emergence" as referring to the reduction in the number of larvae that successfully develop into adult insects has been incorporated into the manuscript for enhanced clarity
313 Is "remarkable" the correct word choice?
The phrase " remarkable " has been rewritten for clarity in the manuscript
316 "That not correlated" is unclear
The phrase "that not correlated" has been rewritten for clarity in the manuscript
Missing Please somewhere in the manuscript (discussion or intro) define the difference types of trichomes your were observing and the function of each.
The introduction now includes a detailed explanation of the different types of trichomes observed in our study
Line 351. Where did you acquire the Gb03 culture?
The GB03 culture was acquired from the microbial collection of Lab 11 in the Department of Molecular Biology at the Faculty of Exact, Physical-Chemical, and Natural Sciences, National University of Río Cuarto
Line 365 How did you separate instars?
The separation of instars in our study was primarily conducted based on size, which is a common method for distinguishing between different developmental stages of larvae due to the significant size differences that typically characterize each instar.
LIne 383. When you did your insect bioassays, how did you ensure that insect feeding over 48 hours was consistent across your plants? Some insects may not have fed at all while others may have consumed the whole plant. Did you have a threshold for damage? Did you correlate damage on the plants with EO or VOC production?
For the insect bioassays, we standardized the feeding conditions by exposing each plant to three previously starved S. frugiperda larvae for a 4-hour period within entomological cages. This controlled exposure aimed to induce an approximate 30% leaf damage, providing a consistent level of herbivory across the experimental plants. While the methodology was designed to promote uniform feeding, the inherent variability in insect feeding behavior means that some variation in the extent of damage between plants is inevitable. However, the study did not specify a precise threshold for damage in relation to EO or VOC production. Future studies could benefit from correlating the extent of herbivore-induced damage with changes in EO or VOC production to further elucidate the relationship between herbivory and plant defense mechanisms
Reviewer 2 Report
Comments and Suggestions for Authors
Dear Authors,
I had review the MS entitled “Simultaneous Impact of
Rhizobacteria Inoculation and Leaf- 2 Chewing Insect Herbivory on Essential Oil
Production and 3 VOCs Emission in Ocimum basilicum”. After peer review the MS, it was
found to be suitable for publication after the correction of the Major mistakes. Authors need to
rectify the MS as per the comment given below:
1. In abstract part, line No. 25- 28, mentioned specific results in brief.
2. In line 29, remove the word EO and write the full form of VOC, PGPR
3. Reference is missing from line No. 33- 36 of introduction part.
4. In line No. 38-39, write about detail of chemical constitute found in recent study of O.
bacilicum
5. As authors have performed the activity in detail, the attachment of experimental images
in MS is highly encouraged (For example; plant with larvae infected etc.) Similarly, as
mention in the trichome density under various treatments, images of
trichrome, would be highly encourage in the MS.
6. In line No. 257-258, write in detail about the specific reason why the S. frugiperda
increase the EO and its major constitutens in O. bacilicum under the investigation.
7. In the line No. 278, write the detail findings of the research comparing
with recent published data.
8. From the line No. 336- 348; make the paragraph short.
9. In line No. 393 use the abbreviation of EO and is full form once only throughout the MS
and other words and abbreviation also. Similarly write the full form of ANOVA.
10. In conclusion part, write about the practical application of your research in detail as well
as future prospective of this research. Avoid the repetition of research findings which is
already discussed in discussion part.
11. The format of the references is not uniform. Make it uniform.
Comments on the Quality of English LanguageNeed revision
Author Response
Review 2
Comments and Suggestions for Authors
Dear Authors,
I had review the MS entitled “Simultaneous Impact of
Rhizobacteria Inoculation and Leaf- 2 Chewing Insect Herbivory on Essential Oil
Production and 3 VOCs Emission in Ocimum basilicum”. After peer review the MS, it was
found to be suitable for publication after the correction of the Major mistakes. Authors need to
rectify the MS as per the comment given below:
- In abstract part, line No. 25- 28, mentioned specific results in brief.
The abstract has been revised to include a brief overview of specific results
- In line 29, remove the word EO and write the full form of VOC, PGPR
The suggested changes have been implemented as requested
- Reference is missing from line No. 33- 36 of introduction part.
The missing reference has been added to the introduction section,
- In line No. 38-39, write about detail of chemical constitute found in recent study of O.
Bacilicum
In the introduction, detailed information on the chemical constituents found in recent studies of Ocimum basilicum has been added,
- As authors have performed the activity in detail, the attachment of experimental images
in MS is highly encouraged (For example; plant with larvae infected etc.) Similarly, as
mention in the trichome density under various treatments, images of
trichrome, would be highly encourage in the MS.
Images of the trichome density under various treatments have been added to the manuscript.
- In line No. 257-258, write in detail about the specific reason why the S. frugiperda
increase the EO and its major constitutens in O. bacilicum under the investigation.
The requested information regarding the specific reasons why S. frugiperda increases the EO and its major constituents in O. basilicum has been added
- In the line No. 278, write the detail findings of the research comparing
with recent published data.
The detailed findings of our research, in comparison with recent published data, have been added to the manuscript.
- From the line No. 336- 348; make the paragraph short.
The paragraph has been shortened as requested
- In line No. 393 use the abbreviation of EO and is full form once only throughout the MS
and other words and abbreviation also. Similarly write the full form of ANOVA.
The manuscript was reviewed, and it was ensured that the full form of each abbreviation was written only once, followed by the use of abbreviations throughout the document.
- In conclusion part, write about the practical application of your research in detail as well
as future prospective of this research. Avoid the repetition of research findings which is
already discussed in discussion part.
The conclusion section has been rewritten to include detailed practical applications and future perspectives of our research
- The format of the references is not uniform. Make it uniform.
We apologize for the inconsistency in the reference format. The entire reference list has now been thoroughly reviewed.
Round 2
Reviewer 2 Report
Comments and Suggestions for Authors
All the comments have been resolved properly. Now the MS is ready for publication.
Comments on the Quality of English LanguageSome minor corrections are required.
Author Response
Review 2
1.Comments on the Quality of English Language: Some minor corrections are required
The manuscript was meticulously revised by a native speaker to improve its language and style